# A Comprehensive Literature Review on the Therapeutic Potential of Platelet-Rich Plasma for Diabetic Foot Management: Insights from a Case of a Neglected Deep Plantar Abscess

**DOI:** 10.3390/healthcare13101130

**Published:** 2025-05-13

**Authors:** Stefania-Mihaela Riza, Andrei-Ludovic Porosnicu, Ruxandra Diana Sinescu

**Affiliations:** 1Department of Plastic Surgery and Reconstructive Microsurgery, Carol Davila University of Medicine and Pharmacy, 050474 Bucharest, Romania; stefania-mihaela.riza@drd.umfcd.ro (S.-M.R.); ruxandra.sinescu@umfcd.ro (R.D.S.); 2Department of Plastic Surgery and Reconstructive Microsurgery, Elias Emergency University Hospital, 011461 Bucharest, Romania

**Keywords:** chronic wounds, diabetic foot ulcers (DFUs), platelet-rich plasma (PRP), regenerative medicine, limb salvage surgery, growth factors, angiogenesis, re-epithelialization, wound healing, tissue regeneration, chronic wound management, advanced wound therapy

## Abstract

**Background**: Diabetic foot ulcers (DFUs) remain a major complication of diabetes, characterized by impaired wound healing, high infection risk, and an increased likelihood of limb amputation. Platelet-rich plasma (PRP) has emerged as a promising adjunctive therapy due to its regenerative properties, promoting angiogenesis, modulating inflammation, and accelerating tissue repair. **Methods**: This literature review explores the current evidence regarding the use of PRP in the management of DFUs. It was conducted using the PubMed database to evaluate the efficacy of PRP in DFUs. The search was restricted to studies published in the last 10 years, including randomized controlled trials, meta-analyses, and systematic reviews. The inclusion criteria focused on studies assessing PRP as a standalone treatment or in combination with other wound care strategies, evaluating key clinical outcomes such as wound healing rates, infection control, tissue regeneration, and amputation prevention. **Results**: A total of 35 studies met the inclusion criteria, including 11 meta-analyses, 15 review articles, and 9 clinical trials. PRP demonstrated potential benefits in accelerating wound healing, reducing inflammation, and promoting granulation tissue formation. Additionally, PRP combined with negative-pressure wound therapy (NPWT) showed superior outcomes in reducing amputation rates. However, findings varied based on patient characteristics, PRP preparation techniques, and treatment protocols. **Conclusions**: PRP represents a valuable adjunct in DFU management, contributing to improved healing outcomes and reduced complications. However, the lack of standardized protocols and variability in clinical results highlight the need for further large-scale, multicenter studies to establish its definitive role in diabetic wound care.

## 1. Introduction

Normal wound healing is a complex and well-organized process which involves cell migration, cell proliferation, and extracellular matrix deposition. However, chronic illnesses such as diabetes can lead to the dysregulation of cellular and molecular signals during this process [1].

Skin and soft tissue infections are common chronic complications of diabetes and may represent a risk factor for lower limb amputation in diabetic patients [2]. Foot infections are particularly prevalent among individuals with diabetes, ranging from superficial paronychia to deep infections involving the bones [3].

The socioeconomic burden of DFUs is substantial, encompassing both direct healthcare costs and indirect losses related to reduced mobility, productivity, and quality of life. It is estimated that up to 20% of diabetic patients with foot ulcers will eventually require some level of amputation, making prevention and effective treatment strategies critical [3,4].

Treatment requires multiple surgical debridements of the infected tissue. In some cases, this approach may involve amputation of the segment at various levels. The debridement of necrotic or infected tissue must be performed with maximum preservation of healthy surrounding structures [5,6]. Following the resolution of the infectious process, patients are often left with complex soft tissue defects. Reconstructive surgery is challenging in these patients due to delayed wound healing. In addition to classic surgical methods, new therapeutic options to stimulate wound healing are being studied [7].

The use of platelet-rich plasma (PRP) is thought to be beneficial for these patients. PRP injection is a common procedure in esthetic and plastic surgery. The product is easily obtainable, and its injection is minimally invasive. Platelets secrete several growth factors that interact with the local environment to promote cell differentiation and proliferation, resulting in re-epithelization and angiogenesis [8].

Chronic infected ulcers present a significant clinical challenge due to impaired healing dynamics, which are characterized by persistent inflammation, elevated bacterial load, and insufficient vascularization. Platelet-rich plasma (PRP), an autologous blood derived product with a high content of platelets, has emerged as a promising therapeutic approach. PRP contains a high concentration of growth factors and cytokines that play critical roles in the wound healing process, including the modulation of inflammation, the promotion of angiogenesis, and the stimulation of cellular proliferation. Cellular studies have demonstrated that PRP significantly enhances the proliferation of fibroblasts and keratinocytes, both essential for extracellular matrix deposition and re-epithelialization [9,10]. In diabetic rat models, PRP has been shown to accelerate wound healing by promoting angiogenesis and collagen deposition [11].

This paper presents a literature review on the potential role of PRP in diabetic foot ulcer management, supported by insights from a representative clinical case.

## 2. Materials and Methods

The literature search was conducted through the PubMed database, with the addition of filters to limit the results to relevant studies for the use of PRP in diabetic foot lesions. These filters include Meta-Analyses, Randomized Controlled Trials, Reviews, and Systematic Reviews, with a restriction to human studies only. This approach ensures that the review synthesizes the most relevant and methodologically rigorous evidence available on PRP in diabetic wound healing. To ensure a high-quality and up-to-date evidence base, only studies published in the last 10 years (the search was limited to studies published between 1 January 2010 and 1 January 2025) and written in English were included.

The following search terms were used individually and in combination: “DFU”, “chronic wounds”, “regenerative treatments”, “PRP”, and “PRP in diabetic ulcers”. The aim was to capture a broad but relevant range of studies addressing the therapeutic application of PRP in the context of diabetic wound management.

All titles and abstracts were screened independently by two reviewers. Full-text articles were then assessed for eligibility. Any discrepancies during the selection process were resolved by consensus discussion or, when necessary, consultation with a third reviewer. To assess methodological quality and transparency in reporting, included studies were evaluated in accordance with the PRISMA guidelines.

To minimize selection bias, studies with both positive and negative outcomes were included. Additionally, this review incorporated studies employing various PRP formulations and combination therapies to reflect the diversity of clinical practice and enhance the external validity of findings.

Inclusion criteria:Studies involving diabetic foot ulcer patients treated with PRP, either as a singular intervention or in combination with other therapies.Clinical outcomes related to wound healing, infection rates, granulation tissue formation, and amputation prevention.

Exclusion criteria:Studies on non-diabetic wounds or other conditions not related to DFUs.Studies with insufficient data on PRP preparation or application.

Outcome measures analyzed: rate and duration of wound healing, reduction in wound dimensions, tissue regeneration and angiogenesis, infection control, and amputation prevention.

This review aims to evaluate the efficacy of platelet-rich plasma (PRP) therapy in the treatment of diabetic foot ulcers (DFUs) using the PICO framework. The participants include patients with DFUs, regardless of ulcer severity. The therapeutic intervention consists of PRP therapy administered as local injections or associated with other wound healing options. The comparison was made with standard wound care approaches, as well as other advanced wound treatments such as negative-pressure wound therapy. The assessed outcomes focus primarily on wound healing parameters, including complete wound closure, time to healing, reduction in wound size, granulation tissue formation, infection rates, and amputation prevention.

## 3. Results

A total of 35 studies met the inclusion criteria, of which 11 were meta-analyses, 15 were review articles, and 9 were clinical trials. All nine such studies included patients with diabetic foot ulcers (DFUs) treated with PRP (gel, injection, or other) (Figure 1).

A.Outcome assessment in clinical trials

We present the main results of the included studies in Table 1 below.

B.Key findings in meta-analyses and reviews

The meta-analyses and systematic reviews consistently highlight the beneficial effects of PRP in diabetic foot ulcer (DFU) healing when used as an adjunct to conventional treatments (Table 2).

C.Limitations and gaps in research

Despite these positive findings, limitations and gaps in the research remain, as highlighted in the reviews:The efficacy of PRP requires further clinical validation, particularly in large-scale, well-designed trials.PRP treatment has a limited effect in certain patient populations, such as those with platelet dysfunction, thrombocytopenia, leukemia, or poor general health status.The systemic metabolic environment in T2DM—including elevated levels of glucose, reactive lipid byproducts, and nitric oxide (NO) metabolites—can lead to PRP cytotoxicity.The clinical application of PRP should be individualized; the etiology of the ulcer, as well as patient-specific factors, should be taken into consideration.The lack of standardized protocols for PRP preparation, concentration, and administration introduces significant variability in clinical outcomes.Many studies do not consistently report key details, such as PRP preparation methods, platelet concentrations, or leukocyte content, making it difficult to compare findings across studies.Differences in centrifugation methods, leukocyte content, platelet concentration, and delivery techniques (injection vs. topical application) make it challenging to compare results across studies.

## 4. Discussion

There is ongoing research on the potential therapeutic effects of platelet-rich plasma (PRP) in the management of diabetic foot complications [8]. It is crucial to bear in mind that not all studies demonstrate unequivocal benefits [17,18,19], and the optimal protocols for PRP application (concentration, frequency, application method) are still a subject of debate [11]. While some studies suggest positive outcomes and accelerated wound healing with PRP [8,13], the results have been controversial [25]. The effectiveness of PRP may depend on various factors, including the patient’s particular status and history, the type of wound, and the PRP preparation method [8].

PRP (platelet-rich plasma) is typically injected into the dermis and the subcutaneous tissue around the wound (Figure 2). In some cases, however, PRP can also be injected into the bed of the wound and the adipose tissue surrounding it, particularly in deeper wounds or areas with significant tissue loss [35].

The concept behind using PRP for the treatment of diabetic foot lesions is to use these regenerative properties to improve wound healing and reduce the complications associated with diabetes [36].

Bacterial biofilms are a major barrier to effective healing in diabetic foot ulcers (DFUs), contributing to chronic inflammation, delayed tissue repair, and resistance to antibiotics and immune clearance. These biofilms create a protective matrix that shields bacteria and impairs healing. Recent research suggests that platelet-rich plasma (PRP) not only supports tissue regeneration but also exhibits antimicrobial and anti-biofilm properties. PRP contains platelet-derived antimicrobial peptides and immunomodulatory factors that can help reduce bacterial load and inflammation at the wound site. Moreover, engineered PRP-based dressings, such as silk protein dual-crosslinked hydrogels combined with PRP or PRP-derived exosomes, have shown enhanced efficacy in diabetic wound models. These biomaterials offer mechanical stability, sustained release of growth factors, and potent antibacterial activity against pathogens like Staphylococcus aureus, E. coli, and Pseudomonas aeruginosa. In bone-mimicking environments, these hydrogels have been shown to disrupt biofilm formation, reduce neutrophil extracellular trap (NET) formation, and improve angiogenesis and re-epithelialization [37].

These findings highlight the potential of advanced PRP-based therapies not only to accelerate healing but also to manage microbial colonization, a critical component in chronic wound pathophysiology. While promising, the transition of these findings to clinical use requires further human studies to validate efficacy, safety, and cost-effectiveness.

A 2025 meta-analysis aimed to compare the therapeutic efficacy of various stem cell groups, PRP, and EGF in the treatment of diabetic foot ulcers (DFUs). Regarding healing time, the PRP group had the shortest mean duration [21]. Recent studies have described PRP-derived exosomes as containing growth factors with accelerated healing properties [38].

Additionally, the majority of the randomized controlled trials (RCTs) that reported the healing time used PRP gel, which is believed to allow a gradual release of growth factors (PDGF, VEGF, TGF-β, etc.) [14,20]. Several other studies have demonstrated that PRP accelerates wound healing and reduces wound size [22,24]. An analysis of 10 RCTs showed that, compared to standard wound care, PRP significantly promotes healing and reduces the healing duration [23].

While numerous studies support the role of PRP in enhancing diabetic foot ulcer (DFU) healing [12,26,27,28,29,30,31,33,34], its impact on major clinical outcomes such as amputation remains less conclusive. For example, a randomized clinical trial by Malekpour Alamdari et al. (2021) involving 90 patients with DFUs reported a significant improvement in wound healing following PRP gel application yet found no statistically significant reduction in amputation rates [16]. This finding highlights the complexity of translating enhanced local healing into broader limb salvage outcomes, particularly in studies with smaller sample sizes or limited follow-up.

Conversely, a large-scale meta-analysis by OuYang H. et al. (2024) [32], which synthesized data from 4826 patients, found that PRP treatment was associated not only with improved wound healing but also with significantly reduced amputation rates and fewer adverse events. They compared the standard of care with negative-pressure wound therapy (NPWT), PRP, hyperbaric oxygen therapy, topical oxygen therapy, acellular dermal matrices (ADMs), and stem cells. Compared to NPWT alone, the combined treatment of PRP and NPWT was more effective in reducing healing time and lowered the incidence of adverse reactions. The study further demonstrated that PRP alone or in combination with NPWT reduces the amputation rate compared to other treatment approaches [39]. This suggests that while individual trials may yield variable results, pooled evidence across diverse populations and treatment settings supports the potential of PRP as a limb-sparing therapy in DFU management.

Given these discrepancies, it is important to interpret findings on amputation with caution. Differences in PRP preparation methods, patient comorbidities, and outcome definitions may contribute to the observed variability. Moreover, as noted in Table 1, some studies did not report amputation outcomes, limiting direct comparisons. These factors underscore the need for further high-quality, standardized clinical trials to clarify PRP’s definitive role in preventing amputations among patients with chronic diabetic wounds.

Diabetes can lead to poor peripheral blood flow and impaired wound healing. PRP may be applied topically or injected into the wound to stimulate tissue regeneration and accelerate the healing process. PRP has been studied for its potential to promote angiogenesis—the formation of new blood vessels—which could improve blood supply to the affected areas [39].

In cases where the patients suffer from obesity, hypertension, and neglected diabetes, infections can worsen due to several connected factors, such as obesity-related inflammation and the impaired immune response caused by diabetes. Elevated glucose levels provide a favorable environment for bacterial growth and biofilm formation. Chronic hyperglycemia also impairs angiogenesis, delaying wound healing and prolonging infections. In patients with neglected comorbidities and poor general health and hygiene status, inadequate glycemic control, delayed medical attention, and inappropriate wound care can lead to colonization by multidrug-resistant organisms, further exacerbating infections [8].

The case presented in Figure 3, Figure 4 and Figure 5 was selected due to the initial severity of a deep plantar abscess in a diabetic patient, which progressed with systemic complications, including pulmonary and cardiac dysfunction. Given the extent of local infection and the patient’s critical condition, major limb amputation was considered a likely outcome.

However, through a multimodal treatment approach combining aggressive surgical debridements, negative-pressure wound therapy (NPWT), and platelet-rich plasma (PRP) infiltration, we achieved effective infection control and progressive wound healing. The integration of regenerative therapy with standard surgical and wound care measures proved instrumental in salvaging the limb and avoiding amputation.

This case underscores the clinical value of incorporating PRP as an adjunct to conventional wound management, especially in complex diabetic foot infections at high risk for systemic deterioration and limb loss.

Chronic inflammation is another common issue in diabetic patients. PRP contains anti-inflammatory cytokines that may help modulate the inflammatory response and create a more favorable environment for healing. PRP has also been explored for its potential to reduce the pain associated with diabetic neuropathy, a common complication of diabetes affecting the nerves [40]. Additionally, it is known that PRP can help modulate symptoms associated with scarring. By promoting healthy healing and minimizing inflammation, PRP can reduce the risk of excessive or pathological scarring, such as hypertrophic or keloids scars. PRP’s anti-inflammatory properties can help reduce discomfort and inflammation associated with scars [41].

To date, there is a lack of clinical studies directly comparing leukocyte-rich PRP (L-PRP) with conventional PRP or leukocyte-poor PRP (P-PRP) in the treatment of chronic wounds in humans. Our review of the current literature indicates that comparative investigations have been conducted almost exclusively in preclinical models, primarily using animal subjects [42,43]. These experimental studies suggest that L-PRP may have superior regenerative properties, particularly through more effective modulation of the inflammatory response and enhanced angiogenesis, especially in the context of pressure ulcer healing [42]. While these findings highlight the potential therapeutic advantages of L-PRP, clinical evidence remains insufficient. Robust, well-designed comparative trials in human populations are needed to establish the relative efficacy and safety of different PRP formulations in chronic wound management.

Despite the promising regenerative potential of platelet-rich plasma (PRP), its therapeutic efficacy in diabetic foot ulcers (DFUs) remains variable. Clinical data suggest that approximately 39% of PRP samples exhibited cytotoxic effects on mesenchymal stem cells (MSCs), with similar inhibitory effects on endothelial cell proliferation [28]. This toxicity appears to correlate with elevated nitric oxide (NO) levels, which have been associated with hyperglycemia-induced oxidative stress in patients with diabetes mellitus. Other studies confirm that increased systemic NO and the accumulation of reactive glucose and lipid metabolites may impair cellular function and viability [44,45].

Multiple factors contribute to PRP inefficacy in T2DM, including the degradation of growth factors by bacterial or host-derived proteases, oxidative damage, and metabolic toxicity. The pathophysiological mechanisms of metabolic syndrome—particularly lipotoxicity and lipoapoptosis—may exacerbate PRP dysfunction. In insulin-resistant and obese individuals, ectopic lipid accumulation in non-adipose tissues leads to mitochondrial dysfunction, endoplasmic reticulum stress, and inflammatory signaling cascades, ultimately impairing cellular repair processes critical for wound healing [46].

Despite these challenges, PRP remains under active investigation as a wound-healing therapy. Efforts to optimize its preparation and preserve bioactive components—such as through biomaterial encapsulation or bioengineered scaffolds—may help overcome current limitations and enhance its clinical utility in chronic wound care [47].

The research is still in its early stages, and more large-scale, well-controlled clinical trials are needed to clearly establish its efficacy and safety. Multicenter studies with larger populations are needed to validate these findings [15,25,48].

## 5. Conclusions

Platelet-rich plasma (PRP) shows promise in the management of diabetic foot ulcers, with evidence suggesting it accelerates wound healing and enhances tissue regeneration. It may help lower the risk of amputation, especially when combined with other therapies like negative-pressure wound therapy.

Despite these benefits, PRP’s effectiveness is inconsistent. In patients with type 2 diabetes, high blood sugar, oxidative stress, and harmful lipid bioproducts can reduce PRP’s healing ability. A major challenge is the lack of standard methods for PRP preparation, concentration, and how it is given, making it hard to compare results across studies. No clinical trials have directly compared leukocyte-rich PRP (L-PRP) with leukocyte-poor PRP (P-PRP) in chronic wounds, though animal studies suggest they may work differently. Also, many studies do not clearly report important outcomes like amputation rates.

To move forward, large, well-designed trials using standardized protocols are needed to better understand how to use PRP effectively and safely in DFU treatment.

## Figures and Tables

**Figure 1 healthcare-13-01130-f001:**
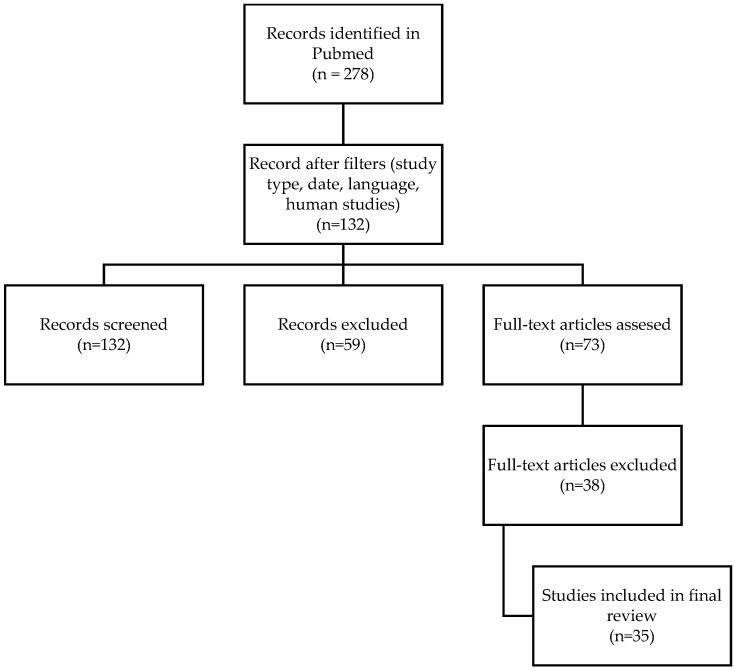
Flowchart illustrating the selection process of studies included in the literature review.

**Figure 2 healthcare-13-01130-f002:**
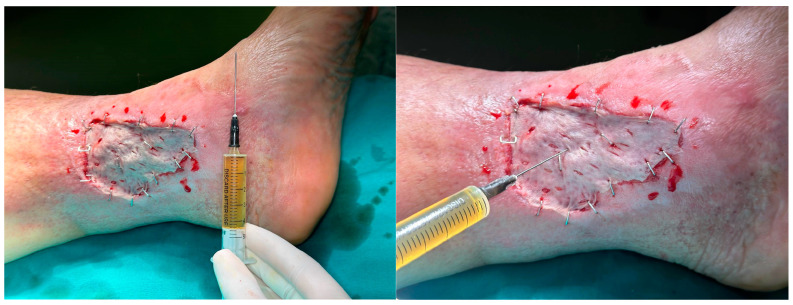
Illustration of PRP injected in our practice.

**Figure 3 healthcare-13-01130-f003:**
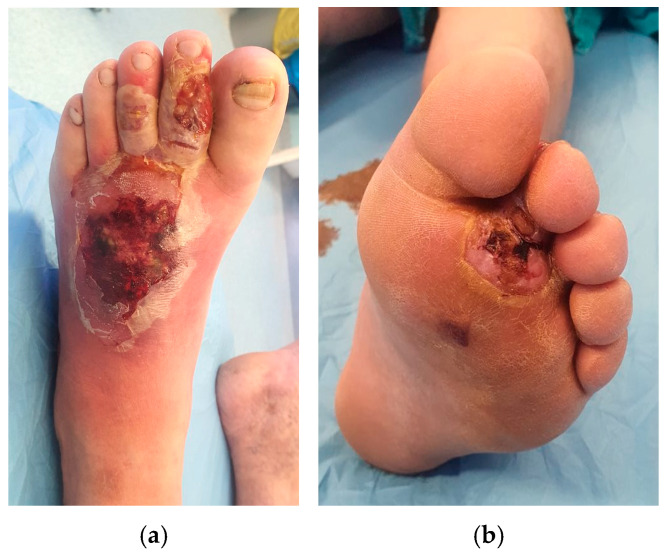
A 58-year-old male with diabetes, obesity, hypertension, and NAFLD presented with a neglected plantar foot abscess, evolving for 14 days after stepping on a toothpick. (**a**) Dorsal aspect; (**b**) plantar aspect.

**Figure 4 healthcare-13-01130-f004:**
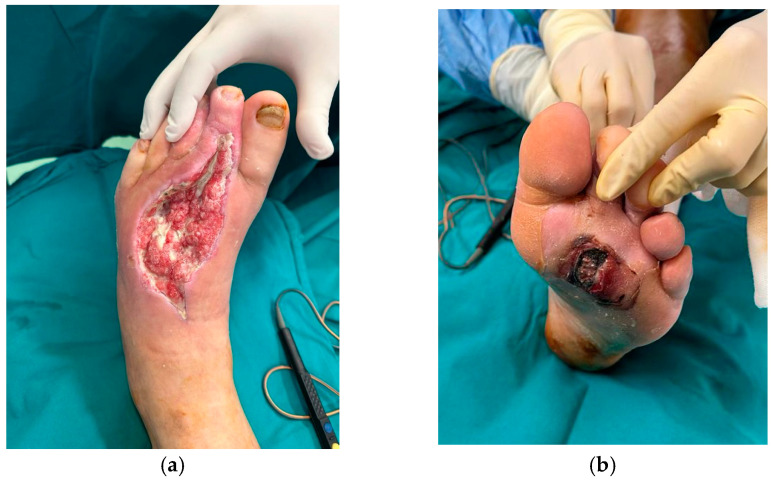
Appearance of the wound 25 days after broad-spectrum antibiotherapy, multiple surgical debridements, and negative-pressure wound therapy. (**a**) Dorsal aspect; (**b**) plantar aspect.

**Figure 5 healthcare-13-01130-f005:**
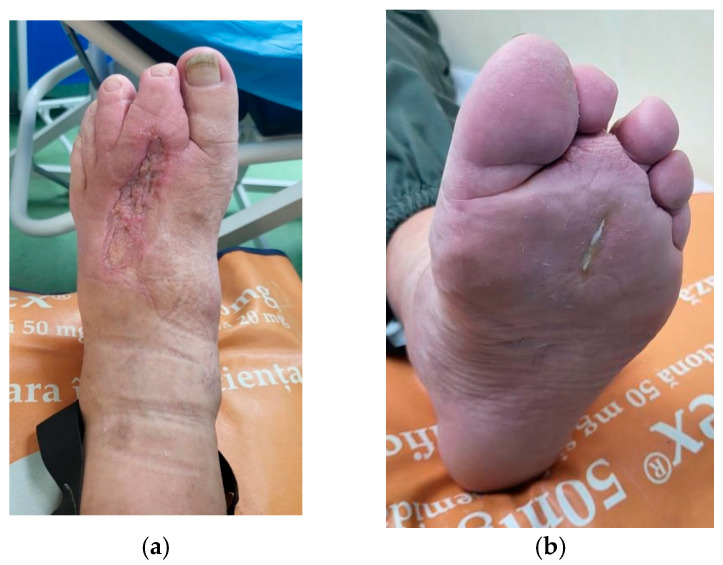
Local appearance of the wound 30 days after skin grafting and two sessions of injectable PRP, administered into the wound bed and surrounding tissue. (**a**) Dorsal view; (**b**) plantar view.

**Table 1 healthcare-13-01130-t001:** Main outcome measures analyzed in reviewed studies (N/A = not available or not recorded in the original source).

Study	No. of Patients (Male/Female)	Mean Age	PRP Technique	Complete Wound Closure	Time to Healing	Reduction in Wound Size	Infection Rates	Amputation Rate
Mohammadi MH et al., 2017 [12]	70 (58/12)	53.8 years	gel	N/A	8.7 weeks	51.90%	N/A	N/A
Singh S. et al., 2018 [13]	55 (34/21)	53.76 years	injections	96.30%	9 weeks	68.80%	N/A	N/A
Elsaid et al., 2020 [14]	12 (8/4)	54.7 years	gel	25%	6.3 weeks	43.20%	N/A	N/A
Smith et al., 2020 [15]	6 (5/1)	57.5 years	injections	33.30%	10 weeks	N/A	N/A	N/A
Malekpour AN. et al., 2021 [16]	47 (30/17)	56.7 years	gel	N/A	7.8 weeks	N/A	N/A	11.6%
Yarahmadi A. et al., 2021 [17]	25 (18/7)	55.62 years	gel	32%	N/A	N/A	N/A	N/A
Hossam EM et al., 2021 [18]	40 (28/12)	54.9 years	injections	95%	6 weeks	≥50%	10%	0%
Alhawari H et al., 2023 [19]	10 (6/4)	61.4 years	injections	90%	12 weeks	N/A	0%	0%
Ohura N et al., 2024 [20]	54	N/A	gel	57.40%	8.1 weeks	72.80%	N/A	N/A

**Table 2 healthcare-13-01130-t002:** Key findings from the reviewed literature on PRP use in diabetic foot ulcers.

Supporting Evidence	Key Findings	Notes
Yang HA et al. (2025) [21]	PRP + conventional treatment accelerates wound healing	Enhanced epithelialization and wound closure observed
Smith J. et al. (2024) [22]		
Peng Y. et al. (2024) [23]		
Su YN et al. (2023) [24]		
Qu W et al. (2021) [25]		
Sridharan K et al. (2018) [26]		
Perussolo J. et al. (2025) [27]	PRP and PRGF reduce wound size and accelerate healing	PRGF also showed additional anti-inflammatory benefits
Izzo P et al. (2023) [28]		
Yin XL et al. (2023) [29]	PRP is effective and safe for DFU healing	No serious adverse effects; favorable safety profile
Vicenti G et al. (2018) [30]		
Gomez PT. et al. (2024) [31]	PRP + standard of care leads to superior wound healing vs. SOC alone	RCT; significant improvement in healing rate in PRP group
OuYang H et al. (2024) [32]	PRP reduces amputation rate and adverse reactions	Notable limb-sparing outcomes in high-risk patients
OuYang H et al. (2024) [32]	PRP + NPWT is more effective for complete ulcer healing vs. SOC	Combination therapy showed synergistic effect on healing
Wong AYW et al. (2024) [33]	PRP reduces ulcer-related adverse events (e.g., infections)	Lower infection rates and improved local wound environment
Deng J et al. (2023) [34]	PRP decreases the risk of amputations	Reported reduction in major amputation rate and prolonged limb salvage

## Data Availability

The original contributions presented in this study are included in the article. Further inquiries can be directed to the corresponding author.

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
