# Peer review of "A Comprehensive Literature Review on the Therapeutic Potential of Platelet-Rich Plasma for Diabetic Foot Management: Insights from a Case of a Neglected Deep Plantar Abscess"

_healthcare, 2025, doi:10.3390/healthcare13101130_

Round 1
Reviewer 1 Report
Comments and Suggestions for Authors
The authors submitted a review entitled A Comprehensive Literature Review on the Therapeutic Potential of PRP for Diabetic Foot Management: Insights from a Case 3 of a Neglected Deep Plantar Abscess for consideration.
The history of studying PRP for the treatment of various pathologies, including DFU, has been going on for several decades. On the one hand, it is promising due to its autologous nature, the richness of growth factors, a kind of ideal cocktail of biologically active compounds created by nature. However, as the authors of the review correctly noted, PRP is difficult to standardize. Autologous nature is both an advantage and a possible risk of inefficiency. For example, in the article [1], the authors demonstrated that some sera from patients with type 2 diabetes even have a toxic effect on the functional activity of mesenchymal stromal cells.
Unfortunately, the review does not discuss the lack of effectiveness of PRP in DFU. The tables generated from the analysis of the literature are interesting, but not sufficient for a full review.
Introduction mainly focuses on the problem of bacterial contamination of chronic wounds. Indeed, bacteria, especially bacterial biofilms, play a major role in the formation of chronic wounds. However, in the main text, in the results, almost nothing is said about this.
The article is more suitable for describing a clinical case. Therefore, it is recommended to rework the article into a description of a clinical case with a discussion, including the proposed literature review, since a full-fledged review lacks discussion and novelty.
1. Solovieva, A.O.; Sitnikova, N.A.; Nimaev, V. V.; Koroleva, E.A.; Manakhov, A.M. PRP of T2DM Patient Immobilized on PCL Nanofibers Stimulate Endothelial Cells Proliferation. Int. J. Mol. Sci. 2023, 24.
Author Response
Comment 1: On the one hand, it is promising due to its autologous nature, the richness of growth factors, a kind of ideal cocktail of biologically active compounds created by nature. However, as the authors of the review correctly noted, PRP is difficult to standardize. Autologous nature is both an advantage and a possible risk of inefficiency. For example, in the article [1], the authors demonstrated that some sera from patients with type 2 diabetes even have a toxic effect on the functional activity of mesenchymal stromal cells.
Response: Thank you for this valuable observation. We have updated the Limitations section to include the findings from the cited article, highlighting the potential cytotoxicity of PRP obtained from patients with type 2 diabetes. Additionally, we expanded the Discussion section to address this concern in more detail, outlining various pathophysiological factors that may impact PRP efficacy. We also referenced additional relevant studies that support this perspective.
Comment 2: Unfortunately, the review does not discuss the lack of effectiveness of PRP in DFU. The tables generated from the analysis of the literature are interesting, but not sufficient for a full review.
Response: Thank you for your insightful comment. In response, we expanded the manuscript to address the lack of consistent effectiveness of PRP in diabetic foot ulcers (DFUs). Building on your suggestion regarding cytotoxicity, we incorporated a discussion on lipotoxicity, the variability in PRP composition, and the lack of standardization in preparation methods. We also noted the limited number of studies comparing leukocyte-rich versus leukocyte-poor PRP in the context of DFU treatment. These aspects have been added to both the Discussion and Limitations sections to provide a more balanced and comprehensive review.
Comment 3: Introduction mainly focuses on the problem of bacterial contamination of chronic wounds. Indeed, bacteria, especially bacterial biofilms, play a major role in the formation of chronic wounds. However, in the main text, in the results, almost nothing is said about this.
Response: Thank you for pointing this out. We have revised the Introduction to better balance the information presented and ensure consistency with the rest of the manuscript. Additionally, we expanded the Discussion section to address the role of bacterial contamination—particularly biofilm formation—in chronic wounds. We also included recent findings on the potential bacteriostatic and bactericidal properties of PRP, to provide a more integrated and comprehensive perspective on its role in wound healing.
Comment 4: The article is more suitable for describing a clinical case. Therefore, it is recommended to rework the article into a description of a clinical case with a discussion, including the proposed literature review, since a full-fledged review lacks discussion and novelty.
Response: Thank you very much for your constructive feedback. We greatly appreciate your comments. In response, we have revised the manuscript to better align with the PRISMA checklist and to strengthen its structure and content as a literature review. Our intention is to present this work as a literature review. However, if the current format is still not in line with the journal’s expectations, we are open to reworking the manuscript into a clinical case report with an extended discussion, as you have kindly suggested.
Reviewer 2 Report
Comments and Suggestions for Authors
This was a very interesting read considering the fact that DFUs are a major health challenge, and PRP use in regenerative medicine is an area immense interest. I applaud the authors for presenting this paper in a structured manner with distinct segments, for effectively utilizing the PICO framework and for presenting appropriate visuals to support the case.
In my opinion, this write-up adds practical insight to the effective use of PRP in conjunction with existing standards of care for treatment/management of complicated wounds. However, could the authors please address the following concerns:
- Provide additional details, if possible, on how the search was conducted to include the exact date range, quality assessment of the included publications, and how article selection bias was controlled for.
- Provide an expanded discussion to include how variations in PRP variation could affect outcomes. For example, there is growing evidence changes in PRP efficiency depending on the type of PRP (leukocyte-rich and leukocyte-poor).
- Some clinical outcomes (infection rate, amputation rate) are missing (why?) yet included in table A and serve as basis for drawing your conclusion.
- There are some contradictory findings where no significant impacts on some clinical outcomes were found when PRP was used. For instance, Nasser Malekpor Alamdari et al published that PRP significantly increased DFU healing regardless of age, gender, smoking and blood pressure status of patients but had no significant impact on the need for amputation. How are the researchers able to conclude that PRP improves healing and reduces amputation rates without addressing the contradictory findings. Furthermore, most of the data on amputation rate (Table A) is missing.
- Could the researcher please refine their table by summarizing PRP characteristics to include platelet concentration and preparation method?
Author Response
Comment 1: Provide additional details, if possible, on how the search was conducted to include the exact date range, quality assessment of the included publications, and how article selection bias was controlled for.
Response: Thank you for your valuable suggestion. We have updated the Materials and Methods section to include the exact date range of the literature search, the criteria for quality assessment of included studies, and the measures taken to minimize selection bias.
Comment 2: Provide an expanded discussion to include how variations in PRP variation could affect outcomes. For example, there is growing evidence changes in PRP efficiency depending on the type of PRP (leukocyte-rich and leukocyte-poor).
Response: To the best of our knowledge and based on our literature review, there are currently no published clinical studies directly comparing leukocyte-rich PRP (L-PRP) with conventional PRP or leukocyte-poor PRP (P-PRP) in the treatment of human chronic wounds. Existing comparative studies are limited to preclinical models, primarily in animals, which suggest that L-PRP may offer superior regenerative effects—particularly through enhanced modulation of inflammation and promotion of angiogenesis in pressure ulcer healing. These findings support the therapeutic potential of L-PRP; however, further clinical research is necessary to validate its comparative efficacy and safety in human chronic wound management.
Comment 3: Some clinical outcomes (infection rate, amputation rate) are missing (why?) yet included in table A and serve as basis for drawing your conclusion.
Response: Some clinical data are marked as unavailable because they were not reported in the original studies. We have updated the relevant section to clearly indicate these instances and to provide clarification where applicable.
Comment 4: There are some contradictory findings where no significant impacts on some clinical outcomes were found when PRP was used. For instance, Nasser Malekpor Alamdari et al published that PRP significantly increased DFU healing regardless of age, gender, smoking and blood pressure status of patients but had no significant impact on the need for amputation. How are the researchers able to conclude that PRP improves healing and reduces amputation rates without addressing the contradictory findings. Furthermore, most of the data on amputation rate (Table A) is missing.
Respone: We appreciate the reviewer’s observation regarding the inconsistent findings related to PRP and amputation rates. Indeed, the randomized clinical trial by Malekpour Alamdari et al. (2021), which included 90 patients with DFUs treated with PRP gel, found no statistically significant difference in amputation rates between the intervention and control groups, despite reporting improved healing outcomes. In contrast, a more recent and comprehensive meta-analysis by OuYang H. et al. (2024), which included data from 4,826 patients with DFUs, concluded that PRP treatment significantly reduced both amputation rates and adverse events compared to standard care. To address this contradiction, we have revised the manuscript to acknowledge the variability in clinical findings and have clarified that although PRP consistently enhances wound healing, its effect on amputation rates may vary depending on study design. Additionally, we have updated Table A to indicate when amputation data was not available in the original studies and have clarified this in the text.
Comment 5: Could the researcher please refine their table by summarizing PRP characteristics to include platelet concentration and preparation method?
Response: We appreciate the reviewer’s suggestion. Indeed, a major limitation in the field is the absence of standardized protocols for PRP preparation, platelet concentration, and delivery methods, which complicates cross-study comparisons. In response to this comment, we note into the text that many studies do not consistently report these details, further highlighting the need for standardized reporting in future research.
Round 2
Reviewer 1 Report
Comments and Suggestions for Authors
The authors' changes to the work have significantly improved it.
Publication in this form is possible.